# Fumagillin, a Mycotoxin of *Aspergillus fumigatus*: Biosynthesis, Biological Activities, Detection, and Applications

**DOI:** 10.3390/toxins12010007

**Published:** 2019-12-20

**Authors:** Xabier Guruceaga, Uxue Perez-Cuesta, Ana Abad-Diaz de Cerio, Oskar Gonzalez, Rosa M. Alonso, Fernando Luis Hernando, Andoni Ramirez-Garcia, Aitor Rementeria

**Affiliations:** 1Fungal and Bacterial Biomics Research Group, Department of Immunology, Microbiology and Parasitology, Faculty of Science and Technology, University of the Basque Country (UPV/EHU), Barrio Sarriena s/n, 48940 Leioa, Spain; xabier.guruceaga@ehu.eus (X.G.); uxue.perezc@ehu.eus (U.P.-C.); ana.abad@ehu.eus (A.A.-D.d.C.); flhernando@ehu.eus (F.L.H.); 2FARMARTEM Group, Department of Analytical Chemistry, Faculty of Science and Technology, University of the Basque Country (UPV/EHU), Barrio Sarriena s/n, 48940 Leioa, Spain; oskar.gonzalezm@ehu.eus (O.G.); rosamaria.alonso@ehu.eus (R.M.A.)

**Keywords:** fumagillin, *Aspergillus fumigatus*, chemical detection, metabolic pathway and regulation, MetAP2 enzyme, cancer treatment, microsporicidal activity, honeybee hive infections

## Abstract

Fumagillin is a mycotoxin produced, above all, by the saprophytic filamentous fungus *Aspergillus fumigatus*. This mold is an opportunistic pathogen that can cause invasive aspergillosis, a disease that has high mortality rates linked to it. Its ability to adapt to environmental stresses through the production of secondary metabolites, including several mycotoxins (gliotoxin, fumagillin, pseurotin A, etc.) also seem to play an important role in causing these infections. Since the discovery of the *A. fumigatus* fumagillin in 1949, many studies have focused on this toxin and in this review we gather all the information currently available. First of all, the structural characteristics of this mycotoxin and the different methods developed for its determination are given in detail. Then, the biosynthetic gene cluster and the metabolic pathway involved in its production and regulation are explained. The activity of fumagillin on its target, the methionine aminopeptidase type 2 (MetAP2) enzyme, and the effects of blocking this enzyme in the host are also described. Finally, the applications that this toxin and its derivatives have in different fields, such as the treatment of cancer and its microsporicidal activity in the treatment of honeybee hive infections with *Nosema* spp., are reviewed. Therefore, this work offers a complete review of all the information currently related to the fumagillin mycotoxin secreted by *A. fumigatus*, important because of its role in the fungal infection process but also because it has many other applications, notably in beekeeping, the treatment of infectious diseases, and in oncology.

## 1. Introduction

*Aspergillus fumigatus* is a saprophytic mold that plays an important ecological role as a decomposer, recycling carbon and nitrogen sources [1,2,3]. It is a ubiquitous fungus with a worldwide distribution, which can be detected in air and soil samples, and even on the International Space Station [4,5,6,7]. This ubiquity is because it is highly adaptive; able to colonize a wide range of environments because of its metabolic diversity, broad stress, and thermal tolerances; and has the ability to spread its conidia easily [4,6,8,9]. In addition, this mold has gone from being considered as just a saprophytic fungus to recognition as one of the most important opportunistic fungal pathogens around, and it is the main causal agent of invasive aspergillosis which has a high mortality rate, between 40% and 90% [2,10,11].

Filamentous fungi produce a remarkable diversity of specialized secondary metabolites (SMs), characterized as bioactive molecules of low molecular weight that are not required for the growth of the organism. Production of these SMs can help fungi in their adaptation to different environmental conditions, improving competitiveness against other microbes or with immune responses during infections [12]. These SMs play diverse ecological roles in fungal defense, communication, and virulence [13], and some of them, owing to their toxic activity, are collectively known as mycotoxins. In recent years, there have been many reviews on the production of theses type of compounds by species from the genus *Aspergillus* [12,14,15,16,17,18,19,20], and specifically, *A. fumigatus* has the potential to produce 226 of these compounds [21]. The genes responsible for the synthesis of SMs are commonly associated with biosynthetic gene clusters [16,22,23] and the *A. fumigatus* genome contains between 26 and 36 putative SMs gene clusters depending on the authors [23,24,25].

Fumagillin is one of these mycotoxins. First isolated from *A. fumigatus* in 1949 [26], it is encoded inside a supercluster on chromosome eight [27,28]. The target of this mycotoxin is the methionine aminopeptidase (MetAP) type 2 enzyme to which it binds and inactivates irreversibly [29]. As MetAPs are essential for the hydrolyzation of the initial methione (iMet) located in the N-terminal of the new proteins being synthesized [30,31], any imbalance produced by MetAP2 inhibition can affect many proteins, some of them implicated in the correct maintenance of cellular safety.

This activity is the basis of the different effects associated with fumagillin. On the one hand, this toxin showed an antibiotic effect as amoebicidal activity inhibiting the growth of *Entamoeba histolytica* [32], and shows similar functions during interaction with macrophages [33]. These studies, among others, led Casadevall et al. [34] to hypothesize that fungal virulence can be based on mechanisms developed to defend against ameboid predators. Besides, fumagillin has pharmaceutical potential for the treatment of microsporidiosis [35], as it is the only effective chemical treatment currently available for nosemiasis caused by the parasitic fungi from the Microsporidia phylum on *Apis* spp. [36]. In fact, it is usually used for the treatment of pests in bee hives [36,37]. However, due to the toxicity of fumagillin, it should be used very carefully and it cannot be used widely. Therefore, less toxic derivatives have been developed to replace fumagillin in some applications. On the other hand, fumagillin has anti-angiogenic activity [29], probably because of its inhibitory activity against the MetAP2 enzyme; consequently, it has valuable pharmaceutical potential and a potential role in the treatment of cancer [30]. Moreover, this toxin is able to inhibit the function of neutrophils [38], inducing cell death in erythrocytes [39] and plays a role in damaging lung epithelial cells which opens the way to fungal invasion [40], perhaps owing to its antiangiogenic properties.

The objective of this review was to collate all current knowledge of this toxin—its chemical characteristics, detection methods, production, metabolic regulation, effects, uses, and its applications in different fields.

## 2. Fumagillin from a Chemical/Analytical Point of View

### 2.1. Fumagillin Physichochemical Properties

Fumagillin (Figure 1) is a small molecule with a molecular weight of 458.54 g·mol^−1^. A decatetraenedioic acid connected to a cyclohexane by an ester bond characterizes its chemical structure. The cyclohexane has also a methoxy group, an epoxide, and an aliphatic chain that derives from a terpene and contains another epoxide. These epoxides are in part responsible for the instability of the molecule.

As a relatively non-polar molecule (predicted partition coefficient, logP: 4.05 [41]), fumagillin has poor water solubility (3.3 mg·L^−1^ [42]) but can be dissolved in organic solvents such as ethanol or DMSO. However, owing to the acidic nature of the carboxylic group (predicted pKa: 4.65 [41]), the polarity of the molecule increases with pH (predicted distribution coefficient, logD: 0.53 at pH 10 [43]). Therefore, the solubility of fumagillin in water is highly dependent on pH.

The acidic portion of the molecule is also a good chromophore responsible for the strong absorption of fumagillin in the ultraviolet range, with maxima around 335 and 350 nm (Figure 2) [44].

### 2.2. Degradation of Fumagillin

In 1954, soon after the discovery of fumagillin [26], a series of three articles on the stability of fumagillin were published: photolytic degradation in alcohol solution [44], photolytic degradation of crystalline fumagillin [45], and thermal degradation in the presence and absence of air [46]. These early papers expressed concerns regarding the degradation of the molecule when exposed to light. In the first publication, Garrett and Eble [44] observed a loss of fumagillin’s activity that they correlated to a decrease in absorptivity at 351 nm. This degradation was attributed to a photolytic degradation caused, primarily, by light below 400 nm. They proposed that the degradation was due to an isomerization process of the tetraenedioic chromophore, and for the first time, coined the term neofumagillin for the degradation products.

Due to the obvious impact of photodegradation on the analytical results, several authors have studied fumagillin stability. Brackett et al. [47] subjected fumagillin to different stress conditions and observed degradation under both strong acidic (1 M HCl) and strong basic (1 M NaOH) conditions. Interestingly, when they studied the impact of UV light exposure, they observed a degradation at 366 nm but not at 254 nm, probably because the molecule does not absorb UV light at that wavelength. They also exposed a fumagillin solution to normal fluorescent laboratory lights and estimated that only around 60% of the analyte remained after six hours of exposure. In a deeper study into the photostability of fumagillin, Kochansky and Nasr [48] exposed 50% ethanol solutions and sugar syrup samples to sunlight and fluorescent light. The former proved to degrade fumagillin faster than the latter and exhibited an initial exponential decay followed by a linear decrease. Dmitrovic and Durden [49] also observed that the degradation of fumagillin in an acetonitrile solution was much faster under sunlight than under a laboratory’s fluorescent light.

The photolytic degradation of fumagillin has also been studied in honey. Assil et al. [50] concluded that after one day of light exposure, only about one-third of the molecule remained. They observed a major degradation product that was analyzed in depth by diode array spectrophotometer, nuclear magnetic resonance (NMR), and infrared spectroscopy (IR). All the results agree with the isomerization process previously proposed by Garret and Eble [44]. In order to reduce the photodegradation of fumagillin the use of amber vials is a useful alternative, as demonstrated by Higes et al. [51].

Although photodegradation is the most important factor related to fumagillin stability, thermal degradation should also be mentioned. For instance, van den Heever et al. [52] paid special attention to this variable in order to study the degradation of fumagillin in bee-hive conditions. They monitored the concentration of fumagillin in honey samples at 21 °C (in the dark and exposed to light) and at 34 °C in darkness (simulated hive conditions). They observed that the fastest degradation occurred at 21 °C under light exposure and more importantly, that the degradation was faster at 34 °C than at 21 °C in the absence of light. The faster degradation at higher temperatures is in accordance with the results obtained by Higes et al. [51] in syrup and in sugar honey patties. In that case, they studied four temperatures (4, 22, 30 and 40 °C) and observed that the degradation rate in syrup increased with temperature, especially in the samples exposed at 40 °C (where no fumagillin was detected after 20 days). On the other hand, fumagillin was more stable in sugar honey patties with losses around 10% after 45 days. Assil et al. [50] also observed that fumagillin in pure honey was moderately stable for 35 days, even at 80 °C. Under the same conditions, water:ethanol solutions were not stable and they attributed this fact to a protective effect of honey, since high sugar concentrations might limit the availability of reactant water.

Considering the degradation processes that fumagillin undergoes, the light induced and thermal decomposition products have been studied for decades [44]. Probably, the most important contributions to the field are the works by Assil et al. [50] using NMR and IR, and by Nozal et al. [53] using mass spectrometry (MS). On the one hand, thermal degradation products have been related to compounds more polar than fumagillin since they elute earlier in reverse phase chromatography. Among these, dihydroxyfumagillin comes from the hydrolysis of the unstable epoxide in the cyclohexane, and fumagillol via a further hydrolysis of the ester. On the other hand, the light induced degradation compounds are slightly more non-polar than the precursor molecule and are related to the aforementioned neofumagillin derived from the isomerization of the decatetraenedioic moiety. Four isomers have been proposed: two cis–trans diastereoisomers and cis–cis and cis–trans isomers (Figure 3).

### 2.3. Analytical Methods for the Determination of Fumagillin

Fumagillin has been widely employed as an antimicrobial agent with honeybees and fish; thus, most analytical methods have been developed for its identification in these matrices. Although these methods are usually only intended for fumagillin’s detection, it has also been analyzed together with bicyclohexylamine [54], a toxic compound used as counter ion in commercial formulations, or together with other residues in honey [55], surface waters [56], and dairy products [57].

Fumagillin has traditionally been analyzed with photometric detection coupled with high performance liquid chromatography (HPLC), using either fixed wavelength instruments or diode array instruments (DAD). More recently, MS based methods have been also developed.

A solid phase extraction (SPE) treatment is commonly applied prior to chromatographic analysis, although this step is not necessary in simple matrices such as fumagillin powder, for which a sample dilution is enough [47]. Direct analysis after sample dilution (10%–20%, w/v) has also been used for honey analysis employing water or water:acetonitrile solutions [47,51], but is not the most common practice. Nowadays, SPE is usually performed after diluting honey with water, typically using polymeric sorbents (Strata-X). After sample loading, the cartridge is cleaned with water: methanol mixtures and fumagillin is eluted with acetonitrile, basified acetonitrile, or acidified acetonitrile depending on the author [49,52,53]. It is remarkable that Kanda et al. [58] proposed a different SPE approach based on the use of weak anion exchange cartridges after QuEChERS extraction.

SPE based sample treatments have also been applied to the analysis of fumagillin in fish tissue, but due to the nature of this matrix, an additional sample treatment step is necessary. Hence, Guyonnet et al. [59] proposed a method which includes a sample extraction with water and acetonitrile prior to SPE. Fekete et al. [60] developed two sample treatments that could be used, depending on the sensitivity requirements. In both of them, acetonitrile was added to tissue and sonicated in order to disrupt the cells. Then, a simple clean-up or an enrichment procedure was proposed using a C18 cartridge in both cases. For the former, the supernatant is passed through the column and directly analyzed. The latter requires a higher amount of tissue and water addition to the supernatant in order to retain fumagillin in the sorbent. Finally, the analyte is eluted with acetonitrile:water (9:1) solution and analyzed by HPLC.

The chromatographic analysis of fumagillin is performed on non-polar stationary phases (C18 or C8), mainly in isocratic mode, although gradient mode has also been applied for some multiresidue or MS methods. Regarding the mobile phase, acetonitrile is the most common organic modifier with the exception of the method developed by Kanda et al. [58], where methanol is employed and also the method proposed by van den Heever et al. [54] where the acetonitrile contains a small amount of methanol (10%). The aqueous mobile phase is usually slightly acidic due to the addition of phosphoric acid [60], formic acid [55], or acetic acid [47,48,50]. Nevertheless, other authors use ammonium formate and formic acid buffers [49,54] or ammonium formate alone to adjust the pH of the mobile phase. The latter was used by Nozal et al. [53] for the determination of fumagillin and is especially interesting because at this pH value (6.5) the carboxylic group is deprotonated. For this reason, the acidic nature of fumagillin was employed by Guyonnet et al. [59] to develop a method based on ion-pairing. A mobile phase at pH 7.8 was used, where tetrabutylammonium acts as a cation to retain fumagillin in a C8 column. To end this section on chromatographic conditions, the comparison of methods proposed by Fekete et al. [61] for the determination of the analyte in fish tissue should be mentioned. This team developed, as far as we know, the only normal phase method for fumagillin using a silica gel column and a complex mobile phase (n-hexane:dichloromethane:dioxane:2-propanol:acetic acid 43:43:9:5:0.1, v/v percent). One of the advantages of the method is that it can be more easily coupled to a liquid-liquid extraction, such as the one the group proposed with dichloromethane:dioxane:2-propanol. Furthermore, it offered complementary selectivity to the normal phase method they developed (which includes bicyclohexylamine as additive in the mobile phase).

The chromatographic separation has traditionally been coupled with photometric detection since fumagillin shows two adsorption maxima in the UV region around 335 and 350 nm (Figure 2). Although the absorption at 335 nm is slightly higher, the 350 nm wavelength has more commonly been used because a higher selectivity is expected [47,50,53,59,60,61]. In the last few years MS based methods have emerged in which fumagillin is analyzed in positive electrospray mode with the exception of the Orbitrap based method developed by Jia et al. [57] in which the [M-H]-ion is used (*m*/*z* 457.22318) for the analysis of dairy products. Nozal et al. [53] used a single quadrupole instrument and measured the protonated molecule (*m*/*z* 459) in selected ion monitoring mode (SIM) for quantitative analysis. Furthermore, they used the potassium adduct (*m*/*z* 497) and a fragment (*m*/*z* 427) for confirmation. All the tandem MS (MS/MS) methods for the analysis of fumagillin use the protonated molecule as the precursor ion but different transitions are monitored for quantitative purposes: 459.2 > 177.0 [52], 459.1 > 233.3 [55], 459.2 > 131 [49], or 459 > 427 [56]. In addition to these quantitation transitions 459.1 > 215.3 and 459.2 > 102.8 transitions have been used for the confirmation analysis.

As expected, MS based analytical methods offer lower limits of quantitation (LOQ) than photometric detection. In this sense, Nozal et al. [53] obtained LOQ levels several times lower with MS compared to photometric detection when they applied a LC-DAD-MS method to honey samples. While the LOQ for the DAD detection at 350 nm ranged from 95 to 150 ng·g^−1^ depending on the origin of the honey, the LOQ for the same honeys ranged from 3 to 10 ng·g^−1^ using SIM detection. Also, 10 ng·g^−1^ was the LOQ achieved by van den Heever et al. in honey using a MS/MS method [54] and the one obtained by Dmitrovic et al. [49] was slightly lower. They calculated the LOQ with different MS/MS instruments and obtained values ranging from 1.1 to 1.6 ng·g^−1^. An order of magnitude lower (0.1 ng·g^−1^) is the LOQ reported by Kanda et al. [58] obtained with a LC-MS/MS method that included sample treatment with QuEChERS and a SPE. Regarding photometric detection, LOQ values reported by different authors in honey and fish range from 5 to 100 ng·g^−1^. Indeed, Fekete et al. [60] proposed two different sample treatments in which 5 ng·g^−1^ LOQ was obtained when enriching the samples and 100 ng·g^−1^ when applying a simple clean-up procedure.

Finally, it must be highlighted that apart from the chromatographic methods described here an enzyme-linked immunosorbent assay (ELISA) method was proposed by Assil et al. [50] in order to detect fumagillin in honey, reaching levels as low as 20 ng·g^−1^.

## 3. Fumagillin from a Biological Point of View

### 3.1. Fumagillin Biosynthetic Gene Cluster

Genetic information for fumagillin production in *A. fumigatus* is found in a supercluster of genes that encodes genes related to the production of three bioactive metabolites, among others, *fumitremorgin B*, *pseurotin A*, and *fumagillin*, which are located on the subtelomeric region of chromosome 8 (Afu8g00100-Afu8g00720) [27,28]. The fumagillin biosynthetic gene cluster, which encodes 15 genes inside this supercluster (from Afu8g00370 to Afu8g00520), is named the *fma* cluster (Figure 3) [28,62]. Unlike other clusters that encode genes for other biosynthetic pathways, an interwoven net between the genes of the pseurotin A and fumagillin clusters has been detected [28]. The nomenclature of the genes involved in this biosynthetic pathway is complex since each research group applied their own terminology. One of them named the genes and enzymes encoded by their enzymatic activity, *fma-PKS* (gene encoding fumagillin biosynthesis polyketide synthase, Afu8g00370) and when the activity was unknown, by the position in the genome of *A. fumigatus* Af293 strain, as *af490* (Afu8g00490) [62,63,64]. Almost simultaneously, another group named the genes in a progression from *fmaA* to *fmaG* using the *A. fumigatus A1163* strain [28]. Table 1 shows the different nomenclatures used and the information collected in different databases. In this review, the open reading frame (ORF), gene, and protein names of NCBI Database [65], *Aspergillus* Genome Database [66], and UniprotKB Database [67] were used, respectively.

Inside the *fma* cluster, there are other genes related to fumagillin production. Among these, the Afu8g00500 gene encodes a putative acetate-CoA ligase, which may be related to the production of the substrates necessary for the enzyme farnesyl synthase to create farnesyl pyrophosphate. However, this has not been proven.

In addition, the gene regulating the production of this mycotoxin (*fumR*) is also found inside this cluster. Dhingra et al. [64] and Wiemann et al. [28] detected this gene and named it *fumR* and *fapR*, respectively. The deletion of this gene brought about a lack of fumagillin and pseurotin A production [28,64].

On the other hand, the genes *Afu8g00430* and *Afu8g00440*, are not related to fumagillin production but are also in this cluster. They encode an EthD domain-containing protein and PsoF (pseurotin biosynthesis protein F) respectively and seem to be related to the production cluster of pseurotin A [28].

Finally, it is worth mentioning that genes encoding two proteins with MetAP activity also belong to this cluster. These genes, similar to the fumagillin target, may be related to the resistance of the fungus to its own toxin, as explained below. One of them, encoded by the *metAP* gene, is a MetAP2-1, and the other one encoded by the gene *Afu8g00460*, is a Putative MetAP type 1 (MetAP1).

### 3.2. Fumagillin Biosynthetic Pathways

The fumagillin metabolic pathway was elucidated by Dhingra et al. [64], Lin et al. [62,63], and Wiemann et al. [28], based on studies carried out with defective mutants of *A. fumigatus*, cloning and obtaining the different enzymes in the pathway in *Saccharomyces*, and performing in vitro studies of its enzymatic activity using intermediaries as substrate. Figure 4 shows the metabolic reactions involved in the pathway.

Briefly, two components are required for the generation of fumagillin, a structure composed of a rearranged highly oxygenated sesquiterpene and a polyketide-derived tetraenoic diacid. They are produced by two different biosynthetic branches [62,63], which are subsequently joined by esterification.

To be specific, one branch of the pathway begins with the conversion of farnesyl pyrophosphate (FPP) to β-trans-bergamotene by the integral membrane-enzyme, Fma-TC, a beta-trans-bergamotene synthase [62]. This intermediate component undergoes three sequential steps of oxidative transformations catalyzed by another membrane enzyme, Fma-P450, a multifunctional cytochrome P450 monooxygenase [63]. In these oxidation steps, β-trans-bergamotene is first hydroxylated at the bridgehead C5 position to yield 5R-hydroxyl-beta-trans-bergamotene; that is subsequently oxidized at the C9 position coupled to cleavage of the cyclobutane C5—C8 bond originating the intermediate component 5-keto-cordycol. Then, the final step of oxidation, an additional epoxidation via this Fma-P450, transforms this compound into 5-keto-demethoxyfumagillol. In Figure 4, these three oxidation steps are shown as a single action. The biosynthetic pathway continues with two sequential transformations, a hydroxylation and subsequent methylation in the C6 position of 5-keto-demethoxyfumagillol to yield 5-keto-fumagillol [63]. Two enzymes participate in these transformations, Fma-C6H, a dioxygenase, and Fma-MT, a polyketide transferase that causes the hydroxylation and methylation, respectively. Lin et al. [63] indicated that Fma-MT was misannotated, as coded by two separated genes, *Afu8g00390* and *Afu8g00400* in the NCBI Database (https://www.ncbi.nlm.nih.gov/), detecting a single joint transcript for these two genes. The protein database UniProtKB [67] has accepted this proposal and these two genes have now been unified as one single protein. The final step consists of a stereoselective reduction of 5-keto-fumagillol by Fma-KR, a stereoselective keto-reductase to obtain fumagillol.

The second branch of the fumagillin biosynthetic pathway produces the dodecapentaenoyl group via Fma-PKS, a fumagillin dodecapentaenoate synthase. This dodecapentaenoyl group is not detected as a free molecule, and Lin et al. [62] have pointed out that it stays attached to the Fma-PKS enzyme. The next step is due the action of Fma-AT, a polyketide transferase that catalyzes the transfer of the synthesized dodecapentaenoyl group onto the fumagillol produced in the other branch, to produce prefumagillin [62]. Finally, the Fma-ABM, converts prefumagillin to fumagillin via an oxidative cleavage with a monooxygenase [63].

### 3.3. Regulation of Fumagillin Cluster

Sensing and responding to environmental clues is critical to the lifestyle of filamentous fungi [73]. In fact, many of the environmental signals that regulate SMs production in *Aspergillus* fungi, including temperature, pH, and carbon or nitrogen sources, also trigger the onset of asexual and sexual development [74]. In spite of this, it is not well understood how environmental variations influence fungi to produce a wide diversity of ecologically important SMs.

During the expression of the genes involved in the synthesis and secretion of SMs, a hierarchical network of master regulators that respond to multiple environmental cues is implied [75] and they are also involved in the complex processes that regulate development in fungi [25]. In fact, to control the transcription of these SMs production-clusters, not only cluster-specific transcription factors, but globally acting transcriptional regulators, are often involved [25].

As pointed out above, the FumR regulates the expression of both the *fma* and the pseurotin clusters [28,64]. In turn, the global regulator of the fungal secondary metabolism, LaeA, controls the supercluster that includes both clusters [17,27,76,77,78,79]. The relevance of LaeA is worth highlighting as it influences the expression of at least 9.5% of the genome (943 of 9,626 genes in *A. fumigatus*), and to be specific, controls the expression of 20%–40% of major classes of SM biosynthesis genes [27]. Moreover, it is part of the “velvet system,” which is a light-dependent system made up by the heterotrimeric complex VelB/VeA/LaeA and is responsible for developmental regulation and the secondary metabolism [73,78]. In this complex, the components do not affect the activation of the gene clusters under different conditions equally [64,73,79] and while the LaeA always regulates the genes positively, the VeA sometimes shows a negative regulation.

In addition to these cellular networks, other signaling pathways related to fumagillin expression, are the G-protein-coupled receptors GprC and GprD [80], the developmental regulators FlbB and FlbE [81], the nonredundant phosphopantetheinyl transferase PptA [82], the catalytic subunit of protein phosphatase Z PpzA [83], the developmental regulators BrlA and WetA [25,84], the VeA-dependent regulator of secondary metabolism MtfA [85], the developmental transcription factor StuA [68], and *MAT1-1* and *MAT1-2* genes of one of the mating-types [86]. All of these signaling networks affect many or even all secondary metabolism clusters, not only the genes involved in fumagillin pathways.

Finally, the host immune response during colonization/infection and the antifungal treatment employed can also activate these secondary metabolism regulators. The activation of the fumagillin cluster during experimental animal infections has been detected, contributing to the invasion and generation of cell damage [40]. Other authors have shown that, in contact with caspofungin, fumagillin [87] and gliotoxin [88] are overexpressed. This implies that the treatment with caspofungin could have adverse effects on host cells during infection. Moreover, taking into account the fact that several fungi can produce echinocandins, such as caspofungin [89], these responses can represent a normal response to environmental signals between fungi [87].

## 4. Biological Activities of Fumagillin

### 4.1. Methionine Aminopeptidases as Cellular Targets of the Mycotoxin

The MetAP is a family of intracellular proteolytic enzymes, originally described as cytosolic proteins [90,91], that fulfils an important role during protein post-transcriptional and co-translational modifications (NH_2_-terminal myristoylation and acetylation among others) [30]. Furthermore, MetAPs are essential for the new proteins because they control the hydrolyzed iMet located in the N-terminal [30,31]. The correct excision of iMet is also necessary to expose a glycine residue where a lipid acid could bind covalently, allowing an efficient association with membranes or other proteins [31]. The maintenance of MetAP activity throughout evolution may have an energy recycling purpose because in lower organisms methionine is the most expensive amino acid to synthesize from an energy efficiency point of view [92].

Broadly speaking, two different types of MetAP enzymes, known as MetAP1 and MetAP2, exist [90,91]. Both types possess distinct substrate specificity but share a similar enzymatic activity. The specificity of MetAPs relies on the second residue of the target protein, and therefore, depending on that amino acid, only one of the isoforms, both, or neither of them, can hydrolyze the N-terminal methionine [93]. Furthermore, while MetAP2 is only responsible for the N-terminal processing of the proteins synthesized de novo, MetAP1 is responsible for processing most of the proteins [94,95]. Structurally, they are quite similar, but the most important difference between them is the helical domain insertion on the surface of MetAP2 [96,97,98]. It is also worth mentioning the fact that both types are not usually expressed simultaneously, Eubacteria expressing only MetAP1, archaebacteria only MetAP2, and eukaryotes both types [99,100,101,102]. An exception is *Synechocystis* sp., which express MetAP1, 2 and a novel type MetAP3 only found in this cyanobacteria [103].

In spite of the fact that in general, only two types of MetAPs are described, there are many isoforms of each enzyme. The classical examples are the six isoforms of MetAP1 that *Arabidopsis thaliana* carries in its genome [104]. Furthermore, each type of MetAP can include different sub-families, such as the human MetAP-1D, also called MAP1D, which is a sub-family of type 1 and is overexpressed in colon cancer [105]. This different distribution and the wide variety of MetAPs point out the complexity of this enzyme group.

Although MetAP1 is very interesting, MetAP2 has been studied more owing to its inhibitory effect on angiogenesis and endothelial cell proliferation. To be precise, the function of MetAP2 related to endothelial cell proliferation was discovered serendipitously when Ingber et al. [106] suffered a fungal contamination in some capillary endothelial cell cultures. It was observed that the fungus, identified as *A. fumigatus* Fresenius, produced a cell rounding gradient zone instead of the normal toxicity produced by other fungal contaminations in cell cultures. Then, they studied the effect of conditioned medium and isolated the active fraction, finally identifying fumagillin as the active compound. This purified fumagillin completely inhibited endothelial cell proliferation, angiogenesis, and tumor-induced neovascularization [106].

However, it was not until 1997 when Sin et al. [29], using TNP-470, a fumagillin derivate, discovered that the fumagillin-binding protein was the MetAP2, which was irreversibly inactivated by the toxin. For that, they used mutant models of *S. cerevisae* null for MetAP1 (∆*map1*) and MetAP2 (∆*map2*) and concluded that because fumagillin inhibited the growth of the ∆*map1* strain, but not the ∆*map2* and the wild type strains, the target of the toxin was the metalloprotease MetAP2, which is highly conserved between humans and fungi [29]. One year later, Liu et al. [107] were the first to define the precise connection between the toxin and MetAP2 as a covalent bond formed between the C3 ring epoxide group of fumagillin and the imidazole nitrogen (Nε2) of the histidine 231 located on the active site of the MetAP2 [107] (Figure 5). The exposure to this mycotoxin inhibits irreversibly the aminopeptidase activity of MetAP2 because fumagillin binds at the action site of the enzyme [29,98].

It must be recognized that in eukaryotic cells, MetAP2 were previously described as a eukaryotic initiation factor-2-associated protein (p^67^), which protects the phosphorylation of the eukaryotic initiation factor-2 alpha (eIF2α) [91,101]. Specifically, the N-terminal lysine of the MetAP2 seems to take part in the protection of eIF2α phosphorylation [111]. Furthermore, p^67^ (MetAP2) can also bind to ERK1 and ERK2, two kinases that have a crucial function in the cell proliferation process [112], inhibiting its phosphorylation [113].

The role of MetAP2 on cell proliferation, and therefore on cancer, has also been widely studied. In fact, several researchers have shown that the proliferation of endothelial cells decreases if MetAP2 is strongly downregulated by an anti-sense oligonucleotide or siRNAs [114,115,116]. Catalano et al. [97], studying mesothelioma cancer cells, discovered that these cells expressed MetAP2 mRNA levels higher than non-cancerous mesothelial cells and that the treatment with fumagillin induces their apoptosis. In contrast, MetAP2 inhibition increased caspase activity, prevented fumagillin-induced apoptosis, but did not affect the telomerase activity of the mesothelioma cancer cells [97]. Overexpression of MetAP2 had been also related with B cells of malignant lymphomas of various subtypes [117], and with cholangiocarcinoma (CAA) cells [116]. In the final case, inactivation of MetAP2 eliminated the proliferation of metastatic CCA [116].

Regarding the MetAPs’ importance to microorganisms, experimental studies demonstrated that *MetAP1* is an essential gene in *Escherichia coli* [118] and *Salmonella typhimurium* [119]. However, in *S. cerevisae*, *MetAP1* and *MetAP2* are only essential when both are silenced [120], affecting the deletion of one of them only to the growth rate [94,120]. In the *A. fumigatus* genome, some genes encode for MetAP2 enzymes, the housekeeping Afu2g01750, and the Afu8g00410 that is encoded inside the fumagillin biosynthetic cluster. The final gene, which encodes MetAP2-1, is expressed together with this mycotoxin when the cluster is activated. Production of this MetAP2-1 enzyme might protect the fungus from the action of its own toxin because it is insensitive to fumagillin activity and/or because the concentration of the target is considerably increased [62,121]. However, it is not yet clear whether this hypothesis is correct.

### 4.2. Effects of Fumagillin Activity on Host Cells

As the MetAP activity is essential for cell viability as explained above [92], fumagillin activity affects not only MetAPs proteins but also all the subsequent activities necessary for the well-functioning of proteins related to cell viability and growth [31].

Among the molecules affected by the inhibition of MetAP, the guanine nucleotide-binding proteins (G protein-coupled receptors or GPCRs) are a ubiquitous and well-studied family of proteins that need to be N-terminally processed [122]. Namely, the myristoylation of Gi/o α subunit allows the association with membranes, indispensable for the efficiency of the protein Gi/o but not for Gs [123]. The pathway starts with an extracellular stimulus caused by the binding of hormones, cytokines, neurotransmitters, growth factors, etc., with its membrane GPCRs characterized by seven membrane-spanning regions [124]. Afterwards, these proteins activate intracellular signaling involved in numerous signal transduction pathways related to development, survival, proliferation, invasion, migration, tumorigenesis, etc. [31,122,123,124,125]. Focusing on the activation of Gi/o, its α subunit can inhibit adenylyl cyclase (AC), incrementing ATP levels and inhibiting protein kinase A (PKA) [124]. At the same time, Giα can pass the signal flow to Giβγ, which regulates phospholipase C (PLC), phosphatidylinositol 3 kinase (PI3K), Ras, and K^+^ channels. PLC stimulates inositol (1,4,5)-trisphosphate (IP3) which mobilizes Ca^2+^ and protein kinase C (PKC) via diacylglycerol (DAG) [126]. In addition, Ras activates the ERK pathway and calcium dependent proliferation. Furthermore, PI3K promotes the cell division control protein Cdc42, responsible for actin reorganization, necessary for cell adhesion, polarization, migration and invasion [127], and the protein kinase B (PKB/AKT), triggering the activation of numerous anti-apoptotic genes (NF-κB, BAD, CREB, etc.) and inhibiting p53 via the activation of FoxO and Mdm2 (Figure 6) [128]. Besides, PI3K activates the extracellular signal regulated protein kinase (ERK) cascade.

Other molecules that must also suffer N-terminal processing, such as Src [129], PKA [130], ARF [131], and calcineurin [132], are also involved in the same processes. Depending on the stimuli, either Gs or Gi/o pathways are activated, and their effects are related to cytoskeletal contraction or relaxation, respectively [124,127]. Together with Cdc42 and G proteins, these signals are able to mediate extracellular matrix cilia formation, phagocytosis, cell adhesion, and cytoskeletal remodeling [133].

Therefore, the activity of fumagillin could affect multiple signaling pathways, so it is easy to understand that there are a large number of different phenotypes and sensitivities produced by fumagillin. For example, an inadequate localization of Gi/o, Src, and Arf proteins produced by the lack of iMet excision means that the cell is unable to produce the transduction signals after the reception of an external stimulus. This would explain the lack of growth in cells exposed to fumagillin when they are treated with insulin-like grow factor (IGF), platelet-derived growth factor (PDGF), or endothelial growth factor (VEGF) [30,134,135]. Moreover, as signal transduction pathways have the objective of rearranging the cytoskeleton via actin polymerization; their alteration may be the reason for failures in the invasion and migration of some cell types [30].

However, the intracellular signaling pathways are not the only metabolic pathways where MetAP2 is involved. For example: thioredoxin (Trx1), cyclophilin A (CypA), Eukaryotic elongation factor (eEF2), and glyceraldehyde 3-phosphate dehydrogenase (GAPDH) are in vivo proven MetAP2 substrates [136].

In addition, the cytostatic and cytotoxic effect observed in fumagillin sensitive cell lines [137] could be due to an increase in reactive oxygen species (ROS). This affects the cell redox state, causing the incorrect functioning of signalization pathways and the lack of thioredoxin antioxidant function [136,137]. In support of this theory is the fact that it has been demonstrated that the addition of N-acetylcysteine, a potent antioxidant, to the medium protected human melanoma tumor cell line B16F10 from TNP-470, induced death by avoiding an increase in ROS [138]. Nevertheless, in neutrophils the exposure to fumagillin inhibits the NADH oxide complex formation by blocking the translocation of the cytosolic component p47^phox^ to the membrane, thus, preventing the production of superoxide. In addition, neutrophils treated with fumagillin show less degranulation and reduced levels of actin filaments, something that could contribute to the inhibition of the structural reorganizations necessary for neutrophil activation [38]. Neutrophils are the first line of defense against conidia reaching the pulmonary alveoli, and therefore, fumagillin can reduce their ability to kill the hyphae or phagocyte conidia of *A. fumigatus* [38].

Another effect attributed to fumagillin is the promotion of eryptosis, which is the suicidal death of erythrocytes by a dysregulation of intracellular calcium concentration [39], which may also contribute to the virulence of *A. fumigatus*. Calcium is regulated by Ras, G proteins, and calcineurin, all of them targets for MetAP2 [122,132,139]. This mechanism significantly increases intracellular calcium concentration, enhances ceramide abundance, triggers phosphatidylserine exposure, and lowers the volume of erythrocytes.

### 4.3. Fumagillin and Other Similar Toxins

The fungal metabolism allows the production of a huge variety of toxins, and that is the case for *A. fumigatus* [1]. In the case of fumagillin, this molecule does not share its effects or structure with other toxins produced by this fungus [140], but there are other organisms that produce similar toxins; for example, Ovalicin, a sesquiterpene produced by *Pseudorotium ovalis*, structurally similar to fumagillin [141]. In fact, they share the same specific target and effect over cells causing same cytostatic and cytotoxic effects [142]. In spite of their being closely related, the mechanism of action of the two toxins has not yet been clarified.

In relation to cellular effect, other toxins are also involved in similar metabolic pathways. Pertussis toxin (PTX), cholera toxin, *Escherichia coli* heat-labile enterotoxin (LT), diphtheria toxin, and *Pseudomonas* exotoxin A are ADP-ribosylating toxins, which do not affect METAP2 activity, but affect the later stages of that signal transduction pathway [143]. Specifically, the A-promoter of PTX catalyzes the ADP-ribosylation of the C-terminal in the α subunits of Gi/o, inhibiting the membrane localization, and therefore, stops signal transduction [144]. Regarding PTX, this also induces an accumulation of cAMP that affects cell polarization, migration, apoptosis, and the response to growth factors [127,143,144]. In the case of the cholera and LT toxins, these enhance AC activity stimulating the Gs pathway, producing the same effect as Gi/o inhibition [145]. Finally, the diphtheria toxin and the *Pseudomonas* exotoxin A have the same mechanism of action, ADP-ribosylation of EF2 producing the inhibition of protein synthesis [146].

Angiogenesis inhibitors or tumor suppressors, Bevacizumab, Dovitinib, Volociximab, etc., focus on suppressing the growth factors’ signal transduction pathways at the membrane, and at the receptor tyrosine kinases level, blocking cytoskeletal reorganization, or interfering with mediator molecules, such as histamine and nitric oxide [147]. These actions are also in accordance with the theories expressed in this review about fumagillin mechanism of action.

## 5. Fumagillin Applications

### 5.1. Angiogenesis and Antitumor Activities

In the early nineties, the antiangiogenic effect of fumagillin was proven. Ingber et al. [106] stated that fumagillin completely inhibits endothelial cell proliferation in the presence of basic fibroblast growth factor (bFGF), and also inhibits tumor suppressor-induced neovascularization in mice [106]. Shortly after, Kusaka et al. [148] proved that fumagillin and its derivative AGM-1470 are antiangiogenic compounds with four different assays. Summarizing their results, locally administered AGM-1470 inhibited the angiogenesis in a chick embryo chorioallantoic membrane assay and in a rat corneal assay. On the other hand, in a rat sponge implantation assay, AGM-1470 inhibited angiogenesis induced by bFGF. Moreover, in a rat blood vessel organ culture assay, both fumagillin and AGM-1470, reduced the formation of blood vessels and the growth of endothelial cells by 90%, although AGM-1470 did so at a lower concentration [148].

As angiogenesis is an important step in tumor and metastasis development, angiogenesis inhibitors have been studied as promising molecules for cancer treatment. In this regard, some fumagillin derivatives have been tested for antitumor effects on tumor cell lines and different animal models, mainly rats and mice, and in only a few human clinical trials. Some of them lowered the growth of human umbilical vein endothelial cells (HUVEC) at low concentrations and caused cell apoptosis at higher concentrations. In an animal model they suppressed tumor growth [149]. Similar effects have also been observed in a CCA cholangiocarcinoma cell line with fumagillin and its derivative TNP-470 [116,150]. In other animal studies with TNP-470, inhibition of tumor growth and lower tumor vascularization were also observed [151]. This compound also prevented an increase in vascular endothelial growth factor (VEGF), hepatocyte growth factor beta (HGF-β), cyclin D, cyclin E, and cyclin-dependent kinase Cdk4 and Cdk3 levels that are present in hepatocarcinogenesis, promoting cell cycle inhibition [152]. On the other hand, the administration of TNP-470 to patients with colon adenocarcinoma reduced liver metastasis in a phase I study [153]. However, the toxicity of these compounds must be taken into account as they can cause serious but reversible side effects, such as encephalopathy, thrombocytopenia, and ataxia [153,154,155].

It has been observed that in murine models of colorectal cancer, fumagillin and its derivative fr-11887 can inhibit colorectal growth and prevent metastasis [156,157]. On the other hand, it has also been demonstrated that the combination of fumagillin derivatives with antitumor molecules like 5-fluorouracil can inhibit the growth of colon adenocarcinoma tumor cell lines [158] or prolong its tumor growth inhibitory effect after the termination of treatment in an experimental murine model [159].

Similar growth inhibitory effects have been also described when TNP-470 was added to pancreatic cancer cell lines cultures. However, although angiogenesis and tumor volume reduction in a murine model of pancreatic tumor after TNP-470 administration was reported, it did not improve animal survival rates [160]. Other studies with TNP-470 treatment reported a reduction in metastatic nodules or tumor dissemination in murine models [161,162,163]. Interestingly, the co-administration of TNP-470 with immune response inducers, such as dendritic cells (DC), has also been studied in a murine model of pancreatic tumors, bringing a reduction in the tumor volume, the density of blood microvessels, and raising the animal survival rate [164]. The role of TNP-470 as an adjuvant in DC tumor vaccines is currently being studied, as it seems that it could stimulate, in vitro and in vivo, the immunogenicity of the DC, and thus, promote polarization of the immune response towards the specific antigen differentiation of T_H_1 lymphocytes [165].

Some authors have also studied the effects of fumagillin on other tumor processes, in such cases as anaplastic thyroid carcinoma, lymphoma, or neuroblastoma in vitro and in vivo, with promising results [30]. Good results with fumagillin derivatives have been observed, in vitro and in vivo, with prostate cancer models [159,166,167], but in humans no conclusive anti-tumor activity was found [168]. In other types of cancer, such as esophageal cancer, we only have some preliminary and unclear results [169].

### 5.2. Fumagillin Antibiotic Uses

#### 5.2.1. Treatment of Microsporidiosis

Intestinal microsporidiosis due to *Enterocytozoon bieneusi* can cause chronic diarrhea, malabsorption, and wasting in immunocompromised patients. It has been demonstrated that fumagillin treatment leads to the clearance of microsporidia and that it could be an effective treatment for chronic *E. bieneusi* infection in immunocompromised patients; for example, those with AIDS [35,170]. However, some patients developed serious adverse events, such as neutropenia and thrombocytopenia, that completely disappeared after the suspension of treatment. Similar results were observed for the treatment of intestinal microsporidiosis with fumagillin in renal transplant recipients [171]. In this case, the secondary effects were not so significant—some abdominal cramps and severe but reversible thrombocytopenia; a decrease in the blood concentration of the immunosuppressor tacrolimus was also observed. It was also reported that fumagillin was a successful treatment for *E. bieneusi* microsporidiosis in two patients with an allogeneic hematopoietic stem cell transplant [172].

Other specialized parasitic fungi from the *Microsporidia phylum* are *Nosema apis* and *Nosema ceranae*. These two parasites can infect *Apis* spp., such as *A. mellifera,* the European bee (also known as the domestic bee), and *A. cerana*, the Asiatic bee. These parasites infect the bee digestive tract midgut epithelial cells and are spread by fecal-oral transmission. The infection lowers worker bee life expectancy, inhibits pollen digestion that leads to insufficient nutrition, raises winter mortality, and lowers honey production [173,174]. The effect of fumagillin on *N. apis* is well known [175,176,177,178]; however, its role against *N. ceranae* has been questioned. Williams et al. [179] suggests that fumagillin is successful at temporarily reducing *N. ceranae*, whereas others suggest that fumagillin is more toxic for honeybees than for *N. ceranae* and those fumagillin low-level residues could lead to hyperproliferation of *Nosema* spp. [180]. However, other authors demonstrated the effectiveness of fumagillin against this parasite [37]. In any case, fumagillin is the only effective chemical treatment currently available for nosemiasis [36]. Nevertheless, it is licensed for use in beekeeping in the United States and Canada but not in Europe, where its use is restricted to special circumstances because of its toxic effects [36,51,181].

#### 5.2.2. Treatment of Other Parasitosis

Another antiparasitic effect of fumagillin has been described. Both fumagillin and TNP-470, potently blocked in vitro growth of *Plasmodium falciparum* and *Leishmania donavani* [182]. Later, it was described that fumagillin and its derivative fumarranol can bind to *P*. *falciparum* MetAP2, inhibiting its growth in vitro and in vivo in a murine model [183]. This opens the door to a possible new treatment for malaria produced by both chloroquine-sensitive and chloroquine resistant strains, and maybe for the treatment of other cryptosporidiosis. In fact, Arico-Muendel et al. [32] described the role of several fumagillin derivatives in inhibiting the in vitro growth of *Entamoeba histolytica*, *P. falciparum*, and *Trypanosoma brucei*. On the other hand, Hillmann et al. [184] described an amoebicidal effect of *A. fumigatus* against the amoeba *Dictyostelium discoideum*, which is abundant in soil, but the effect was principally attributed to gliotoxin by the authors. However, fumagillin had comparably little influence on the viability of amoeba, with a minimum inhibitory concentration 50 (MIC50) towards *D. discoideum* 40 times higher than gliotoxin [184].

### 5.3. Other Applications

The Vpr is a protein that appears on HIV-1 virions. It aids efficient translocation of the proviral DNA into the nucleus and is required for the HIV-1 infection of non-dividing cells, such as macrophages. It is also involved in the activation of viral transcription, induction of cell cycle G2 arrest, and apoptosis of the host cells. For all these reasons, it is a suitable candidate for the development of new treatments based on its inhibition. Some authors have seen that fumagillin can inhibit Vpr-dependent viral gene expression upon the infection of human macrophages [185] and proposed a role of fumagillin as a novel type of AIDS treatment.

On the other hand, it has been described that in early stage development of adipose tissue, adipogenesis is closely associated to angiogenesis [186]. Taking into account the fact that fumagillin is an angiogenesis inhibitor and that weight loss is a common side effect of fumagillin administration in human trials, some authors have studied the role of fumagillin in adipose tissue development. They observed that treatment with fumagillin impaired diet-induced obesity in mice, associated with adipocyte hypotrophy, without any significant effect on adipose tissue angiogenesis [187]. Nevertheless, another study expounded that effects on differentiation of preadipocytes cannot explain the inhibitory effect of fumagillin-like compounds on adipose tissue formation, as they showed only a minor effect on in vivo adipocyte differentiation [188].

## 6. Conclusions

In conclusion, fumagillin seems to be an important factor in promoting rapid adaptation of the fungus *A. fumigatus* to different stresses, including contact with the immune system and lung tissue of the host, favoring infections. As that delay in diagnosis is one of the causes of the high mortality rate associated with invasive aspergillosis, the use of the fumagillin as a biomarker could favor an early diagnosis. The fumagillin activity triggers the inactivation of the enzyme MetAP2, which is of paramount importance for the processing of multiple important proteins. Therefore, this mycotoxin causes damage to the host during infective processes and protects the fungus against environmental aggressions, such as predators. This toxic activity has been used widely by humans in different areas. In fact, due to the inhibitory effect of fumagillin over the angiogenesis and cell proliferation, one of the most promising applications is the control of different types of cancer cells. Likewise, the activity that fumagillin presents against some parasites has allowed its use, for example, in treatment of infections by *Nosema* spp. in bees. Consequently, many analytical methods have been developed for the analysis of fumagillin in several matrices, including honey. The problem that its toxicity brings may be solved through the development of fumagillin derivatives that maintain their activity but with lower levels of toxicity, an endeavour that may generate new alternative treatments for some types of cancers or infections.

## Figures and Tables

**Figure 1 toxins-12-00007-f001:**
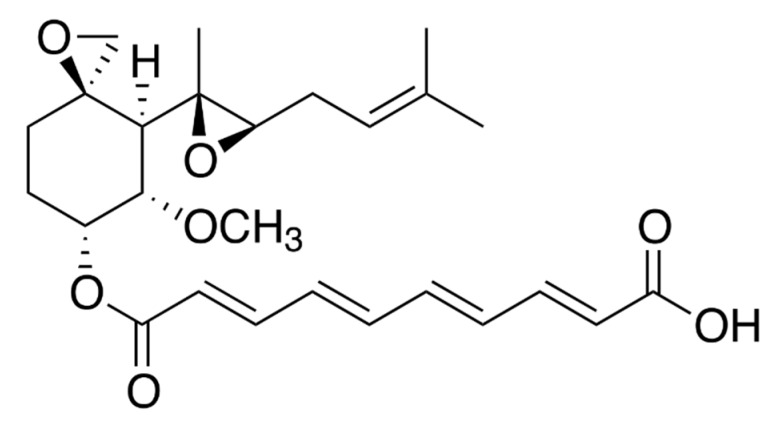
Chemical structure of fumagillin.

**Figure 2 toxins-12-00007-f002:**
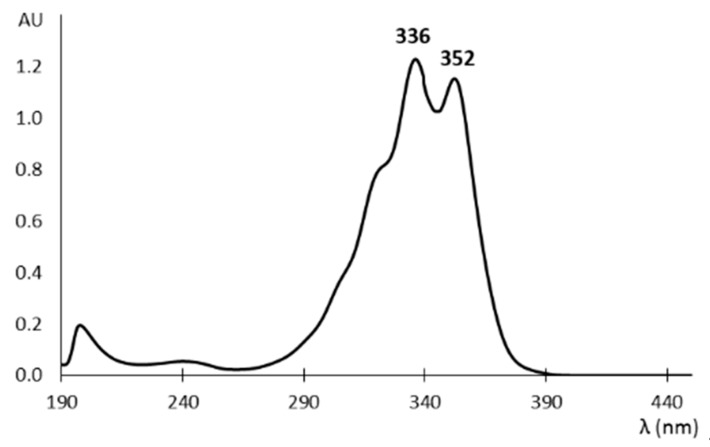
Absorption spectrum of fumagillin solution at pH 2.

**Figure 3 toxins-12-00007-f003:**
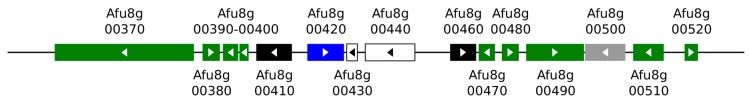
Arrangement of genes in the fumagillin cluster of *A. fumigatus* Af293 strain (Afu8g00370-Afu8g00520; ORF names, Source NCBI Database [65]. *Green*: genes implicated in biosynthetic pathway; *blue*: regulatory gene; *gray and black:* other genes related with fumagillin; *white*: non-related fumagillin genes.

**Figure 4 toxins-12-00007-f004:**
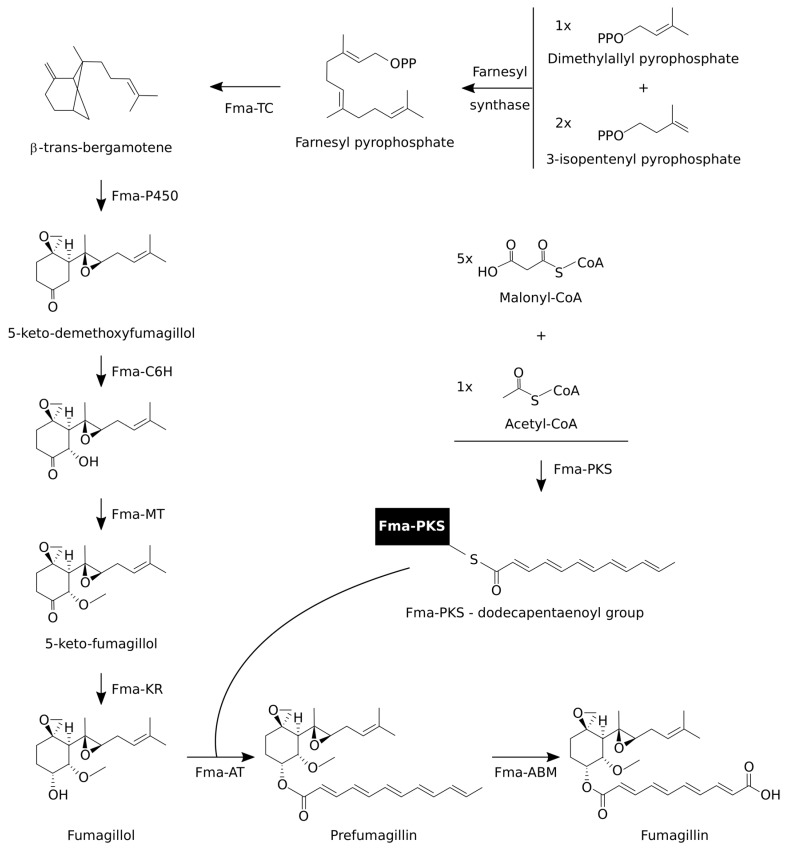
Fumagillin biosynthetic pathway.

**Figure 5 toxins-12-00007-f005:**
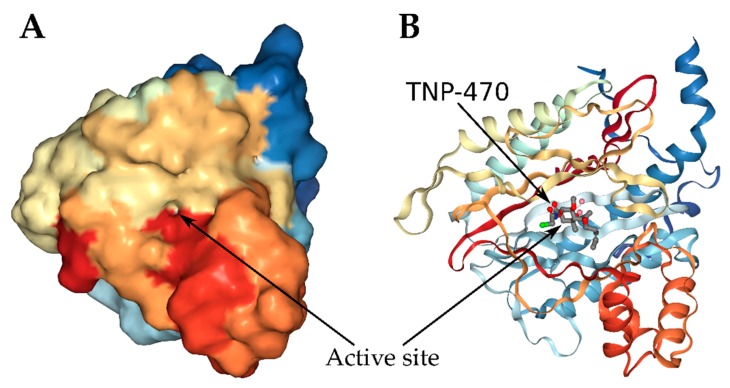
Human MetAP2 3D image. (**A**) Frontal view of enzyme with the active site and (**B**) a view of the MetAP2 chain coupled to TNP-470, a fumagillin derivative in active site. Rainbow colors: blue (N-terminus) to red (C-terminus). TNP-470 view as ball and stick. (PDB ID: 1B6A, NGL Viewer [108] RCSB PDB [109,110]).

**Figure 6 toxins-12-00007-f006:**
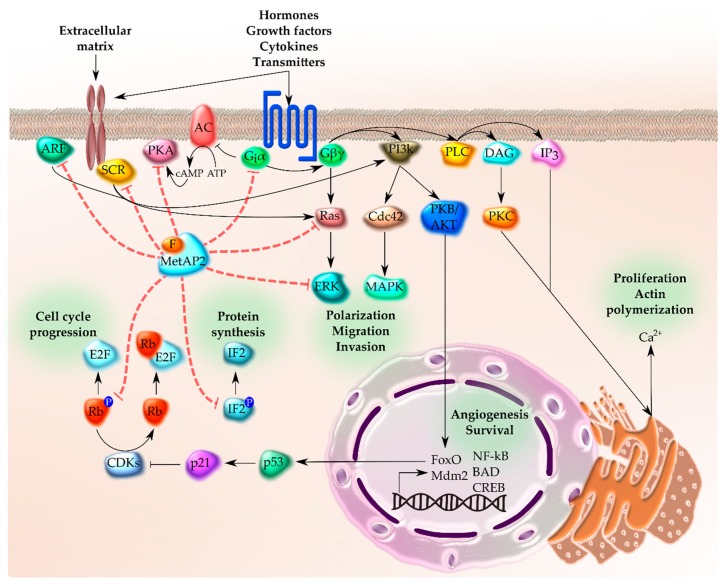
Schematic diagram depicting MetAP2 implication in different pathways. The scheme represents the normal functioning of cascades (black arrows) and the effect of MetAP2 inhibited by fumagillin (red discontinuous arrows). Fumagillin (F) binds covalently to MetAP2 and provokes the inhibition of G protein, SRC, and the ARF signaling pathway at different levels, and the protection of Rb and IF2 against phosphorylation (P).

**Table 1 toxins-12-00007-t001:** Composition of fumagillin biosynthetic cluster.

	Gene Name ^b^	Other Gene Names ^c^	Protein Abbreviated (Complete) Name ^d^	UniProtKB ^d^	Deletion Mutant	Ref.
Afu8g00370	*fma-PKS*	*af370* *fmaB*	Fma-PKS (Fumagillin dodecapentaenoate synthase/Fumagillin biosynthesis polyketide synthase)	Q4WAY3	Fumagillin production abolished Fumagillol production	[28,62]
Afu8g00380	*fma-AT*	*af380* *fmaC*	Fma-AT (Polyketide transferase af380/Fumagillin biosynthesis acyltransferase)	Q4WAY4	Fumagillin production abolished Fumagillol production	[62]
Afu8g00390 /00400	Afu8g00390	*af390-400* *fmaD*	Fma-MT (O-methyltransferase af390-400/Fumagillin biosynthesis methyltransferase)	A0A067Z9B6	Fumagillin production abolished Demethyl-fumagillin production	[28,63]
Afu8g00410	*metAP*	*fpaII*	MAP2-1/MetAP2-1 (Methionine aminopeptidase 2-1/Peptidase M)	Q4WAY7		[68]
Afu8g00420	*fumR*	*fapR*	FumR (C6 finger transcription factor fumR/Fumagillin gene cluster regulator)	Q4WAY8	Fumagillin and pseurotin gene clusters silenced	[28,64]
Afu8g00430	Afu8g00430		(EthD domain-containing protein)	Q4WAY9		[62,69]
Afu8g00440	Afu8g00440	*psoF*	PsoF (Baeyer-Villiger monooxygenase/Dual-functional monooxygenase/methyltransferase psoF/Pseurotin biosynthesis protein F)	Q4WAZ0	Pseurotin production abolished	[28,62,69,70,71]
Afu8g00460	Afu8g00460	*fpaI*	(Methionine aminopeptidase)	Q4WAZ1		[28,62]
Afu8g00470	Afu8g00470	*af470* *fmaE*	Fma-ABM (Monooxygenase af470/Fumagillin biosynthesis antibiotic biosynthesis monooxygenase superfamily monooxygenase)	Q4WAZ2	Fumagillin production abolished Prefumagillin accumulation	[28,62,63]
Afu8g00480	Afu8g00480	*af480* *fmaF*	Fma-C6H (Dioxygenase af480/Fumagillin biosynthesis cluster C-6 hydroxylase)	Q4WAZ3	Fumagillin production abolished 6-demethoxy-fumagillin accumulation	[28,62,63]
Afu8g00490	Afu8g00490	*af490*	Fma-KR (Stereoselective keto-reductase af490/Fumagillin biosynthesis cluster keto-reductase)	Q4WAZ4	Fumagillin production strongly decreased	[62,63,72]
Afu8g00500	Afu8g00500		(Acetate-CoA ligase, putative)	Q4WAZ5		[62,72]
Afu8g00510	Afu8g00510	*af510* *fmaG*	Fma-P450 (Multifunctional cytochrome P450 monooxygenase af510/ Fumagillin biosynthesis cluster P450 monooxygenase)	Q4WAZ6	Fumagillin production abolished, β-trans-bergamotene accumulation	[28,62,63]
Afu8g00520	*fma-TC*	*af520* *fmaA*	Fma-TC (Fumagillin beta-trans-bergamotene synthase af520/Fumagillin biosynthesis terpene cyclase)	M4VQY9	Fumagillin production abolished	[28,62]

^a^ NCBI Database [65], ^b^
*Aspergillus* Genome Database [66]. ^c^ Other name applied in Lin et al. [62], 2014; Wiemann et al. [28]. ^d^ Name of protein and code of the UniProtKB Database [67].

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
