# Peer review of "Fumagillin, a Mycotoxin of Aspergillus fumigatus: Biosynthesis, Biological Activities, Detection, and Applications"

_toxins, 2019, doi:10.3390/toxins12010007_

Round 1
Reviewer 1 Report
Line 18: ...hive infections are... should perhaps be ...hive infections with Nosema spp. are...
Line 150-151: Figure X3 does not exist in the text, and no other figure in the text shows the different isomers as described
Line 153: ...with honey and fish... should be with honey bees and fish...
Line 203: should be Figure 2, and not Figure X2
Line 217: ...times smaller with.. is probably better worded as ...times lower with...
Line 272 should be esterification, and not sterification
Line 351-3 is in boldface (should it be ?)
Author Response
Line 18: ...hive infections are... should perhaps be ...hive infections with Nosema spp. are...
- It has been modified.
Line 150-151: Figure X3 does not exist in the text, and no other figure in the text shows the different isomers as described.
- The numbering of the figure has been corrected to Figure 3.
Line 153: ...with honey and fish... should be with honey bees and fish...
- It has been modified as honeybees.
Line 203: should be Figure 2, and not Figure X2
- It has been corrected to Figure 2.
Line 217: ...times smaller with.. is probably better worded as ...times lower with...
- It has been changed in the text
Line 272 should be esterification, and not sterification
- It has been corrected
Line 351-3 is in boldface (should it be ?)
- Sorry but we did not find the bold text indicated by the reviewer. In our document this text is not in bold.
Reviewer 2 Report
The presented review is comprehensive, well written and clearly organized. As it has been almost 6 years since the last review article on fumagillin, it can be of interest as an updated source of information to a broad range of readers.
Therefore, I would like to recommend it for publication in Toxins with following minor revisions:
1) I found reference to Figure X3 on page 4, lines 150/151 and Figure X2 on page 5, line 203. What do they refer to? There are no figures with tehese numbers in the manuscript.
2) On page 4, lines 123 and 142, number 1 in a supperscript "1" is used twice without any further explanation. What does it mean/refer to?
3) On page 4, line 123, the authors mention an article by Garret and Eble without any reference. Please, provide it in the correct form.
4) The last two paragraphs of the chapter 3.2, lines 301 to 316, would fit more to the chapter 3.1, as they are focused on the fumagillin biosynthetic cluster. Please, consider their relocation.
5) I would suggest a small change in the caption of Table 1., page 7 line 258 - my proposal is "Composition of fumagillin biosynthetic cluster."
6) I would suggest following changes in the caption of Figure 4., page 9 line 300 - my proposal is "Fumagillin biosynthetic pathway (adapted from..."
7) The caption of Figure 5., page 12, lines 403 to 406, is in the format of the main text. Please, use the proper format.
8) In the last paragraph of the chapter 4.1, page 12 lines 423 to 432, all Latin names used for different organisms should be in intalics. This is the same case for the word "Microsporidia" on the page 16, line 587.
9) I would replace phrase "metabolic pathways" on the page 13 line 462 with "regulatory pathways" or "signalling pathways".
10) Please, replace the reference on the page 13 line 485 "(Fallon, 2000)" with the correct reference from.
Author Response
The presented review is comprehensive, well written and clearly organized. As it has been almost 6 years since the last review article on fumagillin, it can be of interest as an updated source of information to a broad range of readers.
Therefore, I would like to recommend it for publication in Toxins with following minor revisions:
1) I found reference to Figure X3 on page 4, lines 150/151 and Figure X2 on page 5, line 203. What do they refer to? There are no figures with tehese numbers in the manuscript.
- The numbering of the figures has been changed to Figure 2 and 3 respectively.
2) On page 4, lines 123 and 142, number 1 in a supperscript "1" is used twice without any further explanation. What does it mean/refer to?
3) On page 4, line 123, the authors mention an article by Garret and Eble without any reference. Please, provide it in the correct form.
- The superscript 1 corresponds to the article of Garret and Ebble, which had the number of reference 44. The 1 superscript has been an error of the automatized references system used. We have corrected it.
4) The last two paragraphs of the chapter 3.2, lines 301 to 316, would fit more to the chapter 3.1, as they are focused on the fumagillin biosynthetic cluster. Please, consider their relocation.
- It has been modified in the text as suggested by the reviewer. The layout of the text has been adjusted to avoid squaring figure 4 on a separate sheet.
5) I would suggest a small change in the caption of Table 1., page 7 line 258 - my proposal is "Composition of fumagillin biosynthetic cluster."
6) I would suggest following changes in the caption of Figure 4., page 9 line 300 - my proposal is "Fumagillin biosynthetic pathway (adapted from..."
- We agree with these suggestions and we have changed the title of Table 1 and the legends of figure 4
7) The caption of Figure 5., page 12, lines 403 to 406, is in the format of the main text. Please, use the proper format.
- This format has been changed
8) In the last paragraph of the chapter 4.1, page 12 lines 423 to 432, all Latin names used for different organisms should be in intalics. This is the same case for the word "Microsporidia" on the page 16, line 587.
- We apologize for the mistake. All Latinized names have been italicized.
9) I would replace phrase "metabolic pathways" on the page 13 line 462 with "regulatory pathways" or "signalling pathways".
- It has been modified in the text as signaling pathways
10) Please, replace the reference on the page 13 line 485 "(Fallon, 2000)" with the correct reference from.
- This is mistake. This reference refers to the article by Fallon et al. 2010, whose number assigned in the references is 38. It has been corrected.